# Improving Spatial Hearing when Wearing Ski Helmets in Order to Increase Safety on Ski Slopes

**DOI:** 10.3390/ijerph192315905

**Published:** 2022-11-29

**Authors:** Josef Seebacher, Markus Posch, Philipp Zelger, Elena Pocecco, Martin Burtscher, Patrick G. Zorowka, Gerhard Ruedl

**Affiliations:** 1Department of Hearing, Speech and Voice Disorders, Medical University of Innsbruck, 6020 Innsbruck, Austria; 2Department of Sport Science, University of Innsbruck, 6020 Innsbruck, Austria

**Keywords:** acoustic horn, skiing, ski helmet, sound source localization, ear pad, winter sports, snowboarding

## Abstract

This study investigated the effect of a new type of ear pads for ski helmets on the hearing performance of 13 young adults (mean age: 22 years). Free-field hearing thresholds and sound localization performance of the subjects were assessed in three conditions: without helmet, with a conventional helmet and with the modified helmet. Results showed that the modified helmet was superior to the conventional helmet in all respects, but did not allow for a performance level observed without a helmet. Considering the importance of precise hearing and sound localization during alpine skiing, acoustically improved ear pads of ski helmets, as demonstrated in this study, can essentially contribute to enhancing the safety on ski slopes.

## 1. Introduction

Recreational alpine skiing is associated with a considerable risk of accidents and injuries. The most common skiing accidents (80–90%) leading to an injury are self-inflicted falls, i.e., falls without the influence of other persons [1,2] (Burtscher et al., 2008, Ruedl et al., 2014). Up to 20% of all injuries resulting from skiing accidents are head injuries [3] (Russell et al., 2010). Head injury risk, however, can be significantly diminished by wearing a ski helmet [3,4,5] (Russell et al., 2010, Ruedl et al., 2011, Shealy et al., 2015). Shealy et al. [5] (2015) found that between winter seasons 1995/1996 and 2011/2012 the incidence of head injuries and of severe head injuries decreased by 62% and by 67%, respectively, while at the same time the rate of helmet usage increased from 8% to 84%.

Nevertheless, helmet usage at skiing still encounters reluctance. A major reason for this may be its adverse effect on sound perception [6] (Ruedl et al., 2012). Evidence shows that wearing a ski helmet significantly worsens hearing thresholds as well as sound source localization in comparison to bare head, and partially even in comparison to wearing a cap [7,8,9] (Tudor et al., 2010, Ruedl et al., 2014, Ruedl et al., 2019). Although the worsening of the hearing thresholds is not much (<20 dB) and is limited to the high frequency region, it may nevertheless have a detrimental impact on a skier’s ability to identify correctly the sound source direction of perceived sounds.

The human auditory system is very skilled in identifying the location of a sound source. In the bare head condition, two sound sources deviating by as little as 2° can be distinguished from each other [10] (Makous et al., 1989). For localizing the source of a sound, the human auditory system uses three cues: (1) interaural differences of arrival times of sounds, (2) interaural differences of sound levels, (3) spectral information generated by the outer ear. The outer ear, especially the pinna, acts like a filter that slightly changes the spectral composition of a sound depending on its incident angle [11] (Blauert et al., 1997). As a result, spectral features of sounds contain information about the position of the sound source, which is evaluated in the central auditory system [12,13] (Middlebrooks et al., 1992, Asano et al., 1990). In particular, the perception of a sound’s elevation and the discrimination of front/rear position of a sound rely on pinna cues [14] (Hofman et al., 1998). Wearing a ski helmet, however, weakens the acoustic function of the outer ear, as the ear pads of the ski helmet are covering the pinna. As indicated above, conventional ski helmets worsen hearing thresholds by about 10 to 20 dB at audiometric frequencies of 3, 4 and 6 kHz [7,8,15] (Tudor et al., 2010, Ruedl et al., 2014, Seebacher et al., 2015).

Remarkably, this frequency region may be specifically important for recognition of sound direction. Blauert et al., have shown that stimuli composed of frequencies between about 2.5 and 6 kHz are rated more likely to come from frontal directions even if their true position is elsewhere [11] (Blauert et al., 1997). The weakened perception of these frequencies when wearing a ski helmet, may explain why a large portion of sound localization errors in the ski helmet condition is associated with front-back confusions [15] (Seebacher et al., 2015). It also substantiates the assumption that the ability to use spectral pinna cues is impaired by the ski helmet because of coverage of the outer ear through its ear pads. A similar effect on hearing thresholds and on sound source localization was found in a study investigating military helmets [16] (Scharine et al., 2015). There, larger errors in sound localization were correlated with greater coverage of the outer ear from the helmet.

Environmental sounds may indicate imminent danger (e.g., objects approaching), hence sound source localization during skiing is crucial in order to react properly. For this purpose, the design of a ski helmet must consider that the coverage of the ears (which is necessary for safety and thermal reasons) should not alter the acoustic function of the outer ear.

In the present study, a new type of ski helmet with acoustically improved ear pads was tested. The new ear pads were self-developed and specifically designed to reduce the distortion of the outer ear’s acoustic function induced by conventional ski helmets. The goal of the study was to quantify the effect of the new type of ear pads on sound source localization. For this reason, sound localization tests were performed in three conditions: with no ski helmet (reference condition), with a conventional ski helmet, and with a ski helmet with modified ear pads. The aim of this study was to verify whether, and to what degree the new type of the ear pad enhances sound localization while wearing a ski helmet, compared to the ear pads of a conventional helmet.

## 2. Materials and Methods

The study was approved by the Institutional Review Board for Ethical Issues of the University of Innsbruck (Certificate of good standing 45/2015, 9 November 2015) and fulfills the ethical standards of the 2013 revision of the Declaration of Helsinki. Prior to participation in the study each subject was informed about the test procedures conducted during the study as well as the risks involved. Participants willing to participate in the study signed a consent form.

### 2.1. Study Questions and Test Conditions

The study questions were:(1)Do the new ear pads compensate the slight hearing loss induced by the ear pads of a conventional ski helmet?(2)Do the new ear pads allow for better sound source localization than do ear pads of a conventional ski helmet?

Study question 1 was investigated by determining the hearing thresholds of subjects. Study question 2 was investigated by measuring their accuracy of sound source localization. For both study questions, the hearing tests were conducted in three conditions:Wearing no ski helmet (reference condition)Wearing a conventional ski helmetWearing a modified ski helmet (i.e., with acoustically improved ear pads).

All hearing tests of the current study were performed under free sound field conditions in an anechoic chamber. Sound source localization was tested with a circular arrangement of 12 loudspeakers mounted in the horizontal plane at the height of the test person’s head. The loudspeakers were equally spaced (in 30° steps) and had a distance of ~1 m to the test person (see Figure 1). Prior to measuring sound localization, a trial run was made to make participants familiar with the testing procedure.

### 2.2. Sample

The participants initially recruited for the study were 20 young adults (10 females, 10 males) with normal hearing. Their age ranged from 19 to 27 years, with an average age of 23.0 ± 2.4 years. All participants underwent otoscopic and audiometric examination before study participation. All of them met the definition of normal hearing according to EN ISO 7029 standards [17] (i.e., hearing thresholds at all frequencies between 0.125 and 8 kHz better than 20 dB HL).

Of the 20 participants, only 13 could be included in the statistical analysis. Seven were excluded because of unreliable sound location judgements. Two of them judged extremely poor (with an unusually high number of front-back confusions in all test conditions) and were considered outliers. The five others judged extremely good, with at maximum two front-back confusions in all ski helmet conditions. For these participants the test conditions were obviously too easy, so that they performed at the high end. Hence, they were also considered outliers. Outliers were removed from the sample in order to prevent the statistical analysis from distortions due to extreme data. Hence, the final sample consisted of 13 persons (5 males, 8 females) with an average age of 22.5 ± 1.9 years.

### 2.3. Design of the Ear Pad

The ski helmet used in this study was of the type Scott Coulter, Model 2016. The shell of the helmet was left unchanged, but its ear pads were modified. The goal of the modification was to minimize sound attenuation and to improve sound directionality without reducing safety and thermal functions of the ear pad. The design of the new ear pad was a result of collaborative research effort of the Department of Hearing, Speech and Voice Disorders, Medical University of Innsbruck, and the Scott Sports company.

After several trials (whose results were not satisfying), the new ear pad was designed in the following way. It has three funnel-shaped openings facing to the front, to the rear, and to the side. Due to their funnel shaping, the openings work like acoustic exponential horns providing not only sound passage, but also sound amplification. An acoustic horn is a passive method of sound amplification, which does not require energy supply, like electronic amplification systems do. The small entrances of the horns are selective to frequencies such that only frequencies above 2 kHz (front opening), 2.8 kHz (rear opening), and 7.5 kHz (sideward opening) are amplified. The cutoff frequencies for the entrances were estimated from the ideal horn equations [18] (Blackstock D, 2000). A prototype pair of this ear pad were manufactured by Scott Sports company and incorporated into one of their ski helmets. This helmet was used in the present study.

### 2.4. Measurement of Hearing Thresholds

Stimuli for measuring hearing thresholds in the free sound field were standardized narrow band noise signals with center frequencies corresponding to audiometric pure tone frequencies 0.25, 0.5, 1, 2, 3, 4, 6 and 8 kHz. They were presented from the frontal loudspeaker of the circular loudspeaker array while the test subject was sitting in the center of the array. All other test conditions were in accordance with the standards of audiometric testing [19] (Böhme & Welzl-Müller, 2004). Calibration and measurement procedures followed audiometric standards EN-ISO8253—1, 2 and 3.

### 2.5. Measurement of Sound Localization

During sound localization tests, the subjects were seated in a height-adjustable chair in the center of the circular loudspeaker arrangement. The height of the seat was adjusted such that the subject’s interaural axis was at the horizontal level of the loudspeaker arrangement. A headrest prevented head movements, and subjects were additionally instructed not to move their head during the test.

Sound presentation was randomized across the array of the twelve loudspeakers. Subjects responded verbally by naming the number of the loudspeaker where they perceived the sound to come from. Numbers of loudspeakers corresponded to the digits of a clock face, with “12” indicating the front position and “6” indicating the rear position [8,9,15] (Ruedl et al., 2014; Seebacher et al., 2015; Ruedl et al., 2019).

As a stimulus, a CCITT (Consultative Committee for International Telephony and Telegraphy) noise burst was used. The CCITT noise is speech modulated noise, which is frequently used in audiometry as a stimulus for directional hearing tests. The stimuli were presented at three different sound levels: at 60, 65, and 70 dB SPL. Presenting stimuli at slightly varied sound levels (called: level roving) aims to interfere with judging a sound’s direction based solely on the head shadow effect. The duration of each stimulus was one second. Presentation of the stimuli occurred in randomized manner, both for sound level and position of loudspeaker. Each stimulus was presented 3 times per loudspeaker, yielding a total of 12 stimuli × 3 levels × 3 conditions = 108 presentations.

### 2.6. Measures and Statistics

Hearing thresholds of subjects were determined in each of the three conditions (no ski helmet, conventional ski helmet, modified ski helmet) were averaged over subjects.

Sound localization accuracy was measured twice: (a) by the angular error and (b) by the rate of front-back confusions. The latter is the percentage of judgements where the position of the stimulus on the frontal axis is judged wrong. The angular error is the deviation of the judged position from the true position of a stimulus. It is computed as the root mean square deviation (in ° azimuth) between judged and true position of the sound source. In the present study, front-back confusion rates and angular errors were computed separately for each of the three conditions and their values were averaged over the whole sample.

Measures of hearing thresholds, angular error and rate of front-back confusions were each analyzed with a Repeated Measures ANOVA with “test condition” as the within-subject factor. The factor had three levels: no ski helmet, conventional ski helmet, and modified ski helmet. Normal distribution and sphericity of the data were tested with Lilliefors test and with Mauchly’s test, respectively. In case of sphericity violation, Greenhouse-Geisser correction to degrees of freedom was applied. Significance level was set at 5%. Post hoc comparisons were made with matched-pair t-tests. For these tests, the significance level was adjusted for multiple comparisons by Bonferroni correction, i.e., 0.05/3 = 0.017.

## 3. Results

### 3.1. Hearing Thresholds

Mean hearing thresholds of the sample in the three test conditions “no ski helmet”, “conventional ski helmet”, and “modified ski helmet” are shown in Figure 2. The conventional ski helmet induced a slight hearing loss in the high-frequency region, with hearing levels worse than 20 dB HL at frequencies 4, 6, and 8 kHz. In contrast, the modified ski helmet had less impact on thresholds: at all frequencies, the levels remained in the range of normal hearing (i.e., less than 20 dB hearing loss). Similarly, the notch in the hearing threshold occurring around 6 kHz was much less prominent with the modified ski helmet than with the conventional one.

Repeated Measures ANOVA of hearing thresholds revealed significance for the within-subject factor “test condition” (F = 200.72; *p* < 0.001). Post hoc pairwise comparisons of the three test conditions showed significant differences (at a Bonferroni-corrected two-tailed significance level of 0.017) between conditions “no ski helmet” and “conventional ski helmet”, between conditions “no ski helmet” and “new ski helmet” and between conditions “conventional ski helmet” and “modified ski helmet”, with all *p*’s < 0.001. The effect of the test condition, indicated by partial eta squared, amounted to 0.91.

### 3.2. Sound Localization: Front-Back Confusions

The median numbers of front-back confusions of the sample in the three different test conditions are shown in Figure 3, left panel. Out of 108 stimulus presentations, the median number of front-back confusions were two in the “no helmet” condition, 10 in the “conventional helmet” condition and six in the “modified helmet” condition.

Repeated Measures ANOVA yielded significance for these differences (F = 5.857; *p* = 0.008). Pairwise comparisons of the three conditions showed significance (at the Bonferroni-corrected significance level of 0.017) for differences between test conditions “no helmet” and “conventional helmet” (*p* = 0.006), but not for differences between conditions “modified ski helmet” and “conventional ski helmet” (*p* = 0.31) and between “no ski helmet” and “modified ski helmet” (*p* = 0.043). This finding indicates that, while the conventional ski helmet significantly worsened front-back judgments compared to wearing no helmet, the modified ski helmet did not so. The effect of the test condition on front-back confusions indicated by partial eta squared amounted to 0.328.

### 3.3. Sound Localization: Angular Error

The medians of the root mean square (RMS) angular error of the sample are shown in Figure 3, right panel, for the three test conditions. The smallest median angular error was observed in the “no ski helmet” condition (24.7°), closely followed from the one in the “modified helmet” condition (25.7°). In the “conventional ski helmet” condition, the median angular error was as large 42.6°.

Repeated Measures ANOVA of angular error revealed significance for the within-subject factor “test condition” (F = 4.939; *p* = 0.016). Post hoc pairwise comparisons of the three test conditions yielded a similar result as with front-back confusions: while the difference between conditions “no ski helmet” and “conventional ski helmet” was significant (*p* = 0.016), differences between other conditions were not. This indicates again that, compared to no ski helmet, a conventional ski helmet significantly increased the sample’s mean angular error, while a modified ski helmet did not so.

## 4. Discussion

The aim of this study was to investigate the “acoustic efficacy” of a new type of modified ear pads of a ski helmet. Conventional ear pads are attenuating the environmental sounds, resulting in a slight hearing loss and in impaired sound source localization while wearing the ski helmet. The new type of ear pads was designed to reduce the degree of sound attenuation and to restore, as far as possible, hearing thresholds and sound localization accuracy. For this purpose, the modified ear pads had three funnel-shaped openings, which worked as acoustic horns that amplify the sound during its passage through the ear pad.

The findings of our study demonstrate that this modification is effective, both with regard to hearing thresholds and to sound source localization. The free-field hearing thresholds of our test subjects improved significantly when their ears were covered with the modified ear pads, compared to conventional ear pads. In addition, the extent of the notch induced by conventional ear pads at frequency 6 kHz, was considerably reduced with the modified ear pads. However, there was still a significant difference to hearing thresholds in the bare head condition: the modified ear pads could not restore the hearing thresholds fully, so that they would correspond to thresholds when wearing no headgear. The differences were nevertheless small (<10 dB) and were limited to frequencies between 4 and 6 kHz.

Improvement of sound source localization was less pronounced, but statistically detectable. Sound source localization was assessed by two measures: the rate of front-back confusions and the angular error. Either measure showed that the accuracy of sound source localization was significantly worsened with the conventional ear pads, but not with the modified ear pads, compared to bare head listening. This finding indicates that the modified ear pads are affecting the accuracy of sound source localization to a lower degree than do conventional ear pads.

Considering the importance of sound perception during alpine skiing, our findings have immense practical relevance. Skiers are usually wearing a headgear, either a cap or a helmet. Both are affecting their sound perception. Although a cap may less degrade the sound localization skills than a ski helmet (Ruzic et al. [20] (2015)), a cap cannot be a substitute for a helmet, as it cannot prevent head injuries in case of an accident. Wearing a ski head, however, should not degrade the skier’s perceptual skills such that his/her risk of an accident is raised. For this purpose, the design of a ski helmet should include acoustically improved ear pads, as proposed by our study.

Remarkably, sound source localization accuracy showed large interindividual variability in the two conditions where a helmet was worn (see boxes and error bars Figure 3). Although wearing the modified ski helmet lead to better average performance, the individual test results in this condition extended over a similar range as in the condition “conventional ski helmet”. In contrast, in the “no ski helmet” condition the interindividual variability was much less. This observation suggests that subjects are differently adapting to wearing a ski helmet. While some are not or only little impaired by the coverage of their ears, others may experience severe perceptual limitations from it. This could lead to increased uncertainty about the locations of sound sources and to degraded sound localization skills when wearing a ski helmet.

Despite its promising findings, our study has several limitations that need to be discussed. The first and most striking one is the study’s purely laboratory character. Hearing performance of test subjects was assessed in free-field sound conditions, with subjects sitting quietly in a chair and all stimuli coming from the same horizontal plane. In real-life skiing, conditions are very different: nor is the subject sitting quietly, nor are sounds so simplified. In fact, environmental sounds are frequently coming simultaneously from different directions, and with different levels and durations, and competing with the background noise from the headwind. In addition, the subject’s (and his/her head’s) movements are affecting the judging of the position of a sound source. In this regard, our findings do not reflect the hearing performance of skiers in real life, but just demonstrate certain acoustic differences between a conventional and a modified ear pad.

Another limitation that must be mentioned is that our testing procedures were possibly too easy. As our entire test subjects were young adults who were experienced in skiing and other kinds of sports, they were also experienced in detecting spatial sounds. Hence, the task of judging the position of stable sound sources was not a challenge to them. This may explain, while a quarter of our sample performed so good that their test results, because of a ceiling effect, could not be used in the statistical analysis. We assume that our testing procedures would have been more selective if test subjects were less experienced in sound source detection. We do not think, however, that this would have substantially altered the results of our study.

## 5. Conclusions

The design of ear pads of ski helmets can easily be modified to improve their acoustic features. Ear pads of conventional ski helmets attenuate sound and through this, cause a degradation of hearing and sound localization acuity. Incorporation of small funnel-shaped openings that act as acoustic horns in the ear pad can mitigate these adverse effects. Ski helmets equipped with modified ear pads enable the wearers to improve their hearing and sound location skills compared to wearing conventional ski helmets.

## Figures and Tables

**Figure 1 ijerph-19-15905-f001:**
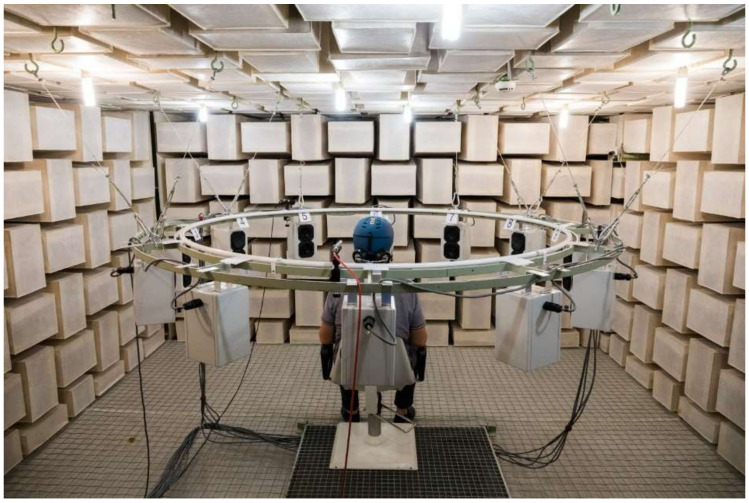
Anechoic chamber: The loudspeaker array of twelve loud speakers and one test person wearing a ski helmet with acoustically improved ear pads are shown.

**Figure 2 ijerph-19-15905-f002:**
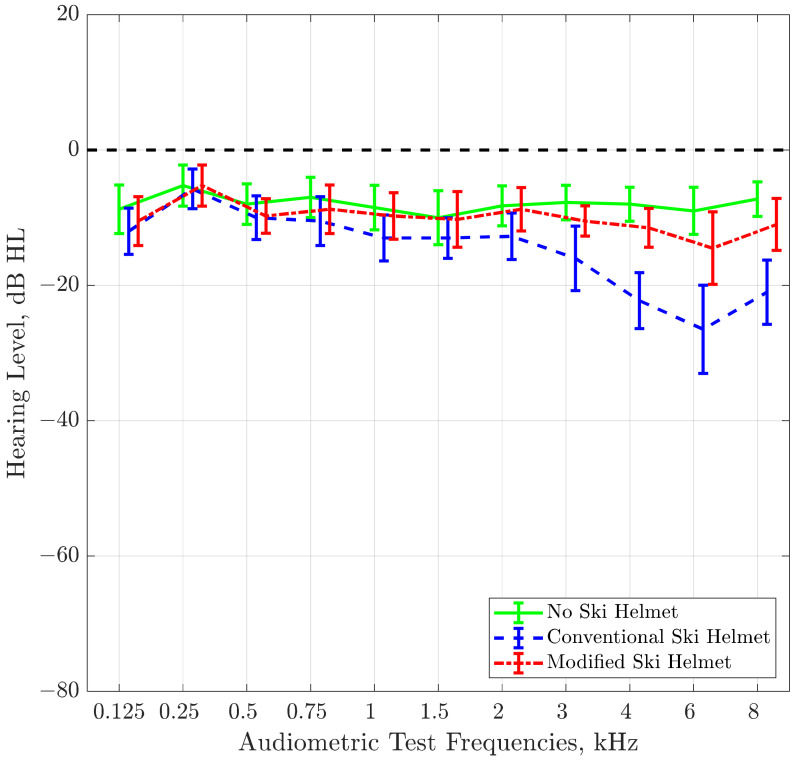
Hearing thresholds (means and standard deviations) of the sample in the free sound field in three test conditions “no ski helmet” (solid line), “conventional ski helmet” (dashed line), and “modified ski helmet” (dotted line).

**Figure 3 ijerph-19-15905-f003:**
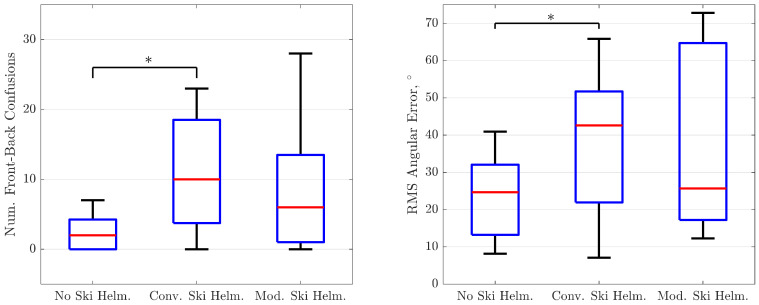
(**Left Panel**) Number of front-back confusions in the conditions “no ski helmet”, “conventional ski helmet” and “modified ski helmet”. Red lines indicate medians, boxes indicate quartiles, error bars indicate first and ninth percentiles. Significant differences between conditions are indicated by a starlet (*). (**Right Panel**) Root mean square angular error of the three test conditions. Symbols same as left panel.

## Data Availability

Not applicable.

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
