# Peer review of "Improving Spatial Hearing when Wearing Ski Helmets in Order to Increase Safety on Ski Slopes"

_ijerph, 2022, doi:10.3390/ijerph192315905_

Round 1

Reviewer 1 Report

General comments

Thank you for author experimental study of a ski helmet based on novel ear pads to improve participants spatial hearing while skiing. The authors focus on an important theme: improving exercise equipment to improve exercise safety.

A significant problem with the current study is the lack of a description of the new ear pad material and the differences in design details compared to traditional ear pads, such as the placement of the two ear pads with the participants external auditory canal. Another major issue is whether the sample size was sufficient; for example, of the 20 initially recruited, 7 were excluded from the experiment. Has the sample size been calculated and shown to be enough to reflect the overall picture?

Detailed comments:

Abstract:
Page 1, lines 10-11:

My suggestion is to change undistorted sound transmission to low attenuation sound transmission to clarify the inevitable hearing loss of ski helmets.

Introduction:

Page 1, lines 30-31:

My suggestion would be to remove the description of the knee injury, because by the numbers it is a bigger problem than a head injury in recreational alpine.

Page 2, lines 68-70:

References are needed.

Materials and Methods:

Page 3, lines 116-120:

To achieve statistical accuracy and reliability, authors should calculate sample size. The subject information is not detailed enough, please add it to the article. It would be nice to have a table with demographics of the participants.

Page 4, lines 145-169:

Necessary supplements should be made to the materials of the new ear pads to further illustrate the material differences between the new ear pads and traditional ear pads, such as whether to use memory materials with slow rebound properties. Meanwhile, to learn more about ear pads, the authors need to show more information about the traditional ear pads and the difference between  traditional and new ear pads.

Page 4, lines 174-175:

Here, the decibel sound pressure level (dB SPL) appears for the first time, and full name is needed.

Results:

Page 6, lines 257:

Please check the number of the figure.

Discussion:

Page 8, lines 349-351:

It is recommended to further discuss the coverage of different ear pads on the auricle, such as the impact of different shielding areas or shielding thicknesses on hearing. Furthermore, although the positions of the ear pads and helmets are standardized, whether the sound source input of the new ear pads is affected by individual differences in the position of the auricles of the subjects. This should be stated to confirm the suitability of the new ear pads.

Page 9, lines 392-399:

The study, conducted under laboratory conditions, demonstrated that the new ear pads improve sound filtering. Meanwhile, in the ski resort scene, ambient noise is unavoidable. Whether the new ear pads based on the three-channel design amplify ambient noise has not been studied, and this should be clearly described. In addition, please give more evidence of researches by others to discuss the results and support your conclusion.

Author Response

Dear Reviewer1

My replies see attached document

Kind regards,

Josef Seebacher

Reviewer 2 Report

Manuscript ID: ijerph-1957407 – Review

Major criticisms:

1. Given the present paper relates to the improvement of spatial hearing, the background literature review is lacking in a great manner.  It is necessary to include the findings of previous studies related to the auditory perception of azimuth. 

2. The present study examined the impact of wearing a ski helmet on the auditory perception.  Yet, no mention is made of previous research that examined the extent to which auditory perception is affected by an individual wearing a helmet.  That research should be included as well, even when it did not investigate skiing helmets.

3. Given the focus of the present study is on auditory spatial perception, the introduction is largely lacking in two respects. First, there is no mention of previous research that examined auditory perception of sound source azimuth. Second, there is no mention of previous research that examined the impact of wearing a helmet on auditory perception. Clearly, those two avenues of research are directly related to the author’s paper and they should be included in the introduction.

4. The end of the introduction should include a brief introduction to the study and the rationale for the study. Doing so would enhance the ability of readers to understand what the study was about and why it was conducted in the manner it was performed.

5. The materials and methods section is in need of restructuring. The information is presented in a disorganized and sometimes confusing manner and often without explanation. Lines 81-83 state that hearing performance was measured in 3 ways. Lines 90-91 state that only 2 tests were performed with the conventional ski helmet. This is contradictory. The methods section should be more coherent.

6. The clarity of the writing is greatly lacking. Too often, the reader is unclear about the point the authors are making and unclear as to why particular manipulations were performed. I highly recommend a thorough review of the manuscript with an emphasis on rephrasing statements and thoroughly explaining both the authors’ thoughts and the rationale for why the study was performed in the manner it was.

7. The results section is confusing. Rather than conduct 2-facotr ANOVAs, the author apparently conducted 1-factor ANOVAs. The former are more appropriate to the design of the study and would yield greater insight into the impact of the various helmet/head conditions. In addition, the analyses are incorrect. Lines 307 – 308, for example, mentions an ANOVA performed using only the head bare data. That is not possible since there would be, as reported in the text, no variable.

Minor criticisms:

1. Introduction, 1st paragraph: Aside from the first sentence which discusses the frequency of skiing as a recreational activity, much the rest of the of the paragraph seems irrelevant to the paper. I would suggest deleting or dramatically reducing the information relating to skiing accidents.

2. Lines 47-48. It is stated that “wearing a ski helmet did not exceed the range of normal hearing.” Then on lines 52-53, it is stated that ski helmets impair hearing thresholds.  This contradiction needs to be clarified.

3. Lines 52-53: A 10-20 dB decrease in hearing frequencies associated with wearing a helmet should not be referred to as a “slight dip”.  Lines 93-94: The hearing impairment resulting from wearing a conventional helmet is referred to as a “mild deficit”. If the impact of the helmet is mild or slight, then one could assume that little to no improvement can be made by using a modified helmet or more acoustically transparent ear pads.  Aside from “slight dip” being a colloquial phrase, it would be better if the author referred to the decrease as significant or not. 

4. Lines 54-58: The point the authors are trying to make is poorly written.

5. Lines 57-58: The statement that timbre influences auditory spatial judgments needs to be supported by empirical research.  Citations should be added.

6. Lines 57-58: The discussion of timbre is unrelated to the argument that manipulations of the pinna influence auditory spatial judgments.

7. Lines 60-61: The statement that a significant number of perceptual errors are front-back confusion needs to be empirically supported. Citations should be added.

8. Lines 68-70: Please provide a citation or support for the statement that the manufacturers of ski helmets are “mainly” interested in thermal properties.

9. Lines 71-72 (as well as lines 52-53): It is stated that individuals wearing a skiing helmet are less able to detect sounds in the range of 5 – 8 kHz. Speech intelligibility is typically limited to 1 and 4 (or 5) kHz. It would be helpful if the authors discussed the importance of the 5 – 8 kHz range in terms of sounds that would be important for skiers.

10. Lines 93 – 96: With regard to the first research question, speech perception and hearing thresholds are independent factors. Observers are capable of accurately detecting speech despite the presence of negatively impacting ear pads. Conversely, it is possible the modified ear pads will not yield the same decrement in hearing thresholds as do conventional ear pads and yet the modified pads may negatively impact speech perception.

11.  Line 102: “Those frequencies”: Based on the information provided at that point in the manuscript, there is no referent to the word “those”. The frequencies impacted by variations in ear pad should be clearly stated and the evidence supporting those frequencies should be provided.

12.  Sample: The exclusion of participants should be performed based on statistical analysis (e.g., more than 2 standard deviations above or below the average for front-back confusions). A more precise and objective metric should be provided to explain the exclusion of individuals.

13. Design of the Ear Pad: Spectrograms of the conventional and modified ear pad should be included in the paper. In addition, pictures of the two pads should be provided.

14.  Lines 199 – 200. Were the 3 sounds levels presented randomly or in blocks?

15. Please review the manuscript for grammatical errors and make the necessary corrections.

16.  Lines 212 – 213: What is the rationalization for amplifying the stimulus by 10 – 15 dB?  Relatedly, why was the sound amplified by a variable amount (10 – 15) and not by a specific amount? More detail needs to be provided in terms of what determined the amount of amplification.

17. Measurement of Speech Perception: What was the task of the participant? How was performance evaluated?

18. Lines 247 – 248: “conventional ski helmet” is listed twice.

19. Results: Pure Tone Thresholds: Statistical analyses are necessary. The difference between the 3 conditions and the changes in thresholds with an increase in test frequency should be evaluated. I would recommend a 3 (head condition) X 11 (test frequency) ANOVA. The results reported in Figure 2 suggest comparisons should also be made between pairs of results: natural vs. modified: 4, 6, and 8 kHz, and modified vs. conventional, 3, 4, 6, and 8 kHz.

20.  Figure 2: The ticks are only sometimes aligned with the audiometric test frequency values.

21. Figure 2: The misalignment of the data points for any given test frequency suggests that different test frequencies were used for the 3 different conditions: bare, conventional, modified.

22.  Figure 2: The scale of the Y-axis (ordinate) should be modified from 20 to -80 dB to 20 to -40 (or -60) dB. As is, the data are compressed and the differences between the 3 conditions are being minimized.

23.  The degrees of freedom reported in the various analyses do not coincide with the experimental design.

24. Sound Localization Bias: Additional statistical analyses are necessary. The difference between the 3 conditions and variations in perception as a function of target azimuth should be evaluated. I would recommend a 3 (head condition) X 12 (azimuth) ANOVA.

25.  Lines 284 – 285: Rather than say the number of front-back confusions was “very low” or how many front-back errors were made, the means and standard deviations should be provided.

26.Line 291: The word “highly” should be omitted.

27.  I would highly recommend the authors report effect sizes and include the results of those analyses in the manuscript.

28. Lines 296 – 297: Based on Figure 3 left panel: It appears there was no significant difference in front-back errors between the conventional helmet and modified helmet.  If this is true, it should also be reported.

Author Response

Dear Reviewer2

My replies see attached document

Kind regards,

Josef Seebacher

Reviewer 3 Report

Review: Seebacher et al. 2022 - Improving spatial hearing when wearing ski helmets in order to increase safety on ski slopes

The authors present a manuscript on spatial hearing with ski helmets. The topic is very interesting and the collaboration between a University Hospital in the middle of the alps and a sports brand is in this context reasonable. The manuscript starts with a good introduction. Unfortunately, the following sections are not as good. Please see my general and detailed comments for details.

General comments:

- sentence structure and use of sub-clauses is reflective of the German language and in some parts overly complex. Some passages are therefore hard to read.

- ‘Participants’, ‘study group’ and ‘test subjects’ are used interchangeably - Confusing structure in the method section - A picture or schematic of the ear pad should be provided

- A picture of the helmet with the positions of the ear pads should be provided - Results section should be improved (see detailed comments)

- Also the discussion section should be improved

Detailed comments:

Line 57: For me “timbre” is more associated with music than environmental sounds. But the use could be more wide spread than I’m used to.

Line 78: Odd structuring and section titles e. g. line 90-92, line 104-111

Line 90-92: Mentioning of different stimuli but the description of the stimuli is found at line 198 – 200 in section ‘Measurement of sound localization’.

Line 104-111: Mentioning of Helmet and ear pads, but there is a hole section on ‘Design of the ear pad’ (line 145)

Line 114: First time mentioning of ‘modified’ ear pads. Line 88 defined the condition with ‘acoustically improved ear pads’. Line 117: ‘physically active’: why is that important?

Line 118: ‘attended the test voluntarily’: odd word choice. Perhaps the authors wanted to say, that the test subjects were not compensated for their time?

Line 121-122: more than 30% of the participants of the study were excluded. 5 of them because they showed low front-rear confusion but were not flagged as outliers like two other excluded participants. There should be a better reason to exclude them.

Line 131-133: The thresholds for normal hearing is defined as hearing level <= 20 dB HL. Certainly not higher than 20 dB. Please add a reference. Line 129-133: The measurement of the hearing threshold is again mentioned at lines 177-184. The two parts should be combined into one section.

Line 130: The cited Norm covers pure tone audiometry via headphones, but the hearings thresholds were measured in free field conditions with loudspeakers, as mentioned in figure 2 (line 268-271). Note I have the ISO-version from 2011. There might be a newer one available.

Line 136-137: exclusion of test subjects because of high front-rear confusion was already mentioned in lines 123-127

Line 171: subsection is called ‘experiment setup and procedure’, but there is no procedure described.

Line 198-200: The CCITT noise is a speech modulated noise, but does not describe a standardized noise burst. The stimulus has to be clearly defined if no citation I given.

Line 205-210: This section seems like groundwork and should be part of the introduction.

Line: 212-213: The text description of the modified signal is inaccurate (10-15 dB) and should be accompanied by a figure of the spectrum. Line 246: What is the degree of coverage of the ear? I Think they mean the measured conditions with the different ear pads

Line 248-251: normality and sphericity were tested and an ANOVA was performed. Why was only the post hoc test done with non-parametric tests, when the data was not normally distributed? The Results only mention the use of ANOVA and paired t-test analysis. Figure 3 shows almost no normally distributed data. Note: ANOVA test can still be performed on data that violates the normality requirement, when the residuals are normally distributed, but the residuals are not mentioned here.

Line 257: Figure 3 does not show the average hearing thresholds

Line 269: straight line  solid line

Line 273: First mentioning of ‘bias’ regarding localization. Should be defined in method section. There is also no plot or table of the results that are analyzed here.

Line 285: text says ‘on average’ and refers to figure 2, but the figure shows the median values.

Line 288-290: very confusing sentence. ‘8 front-rear confusions out of 13 subjects’ implies a total of 8 over all subjects, but in the end it’s the average value for the group. And again, it’s not the average that is shown in figure 3.

Line 294: The text mentions the “head bare” condition, but the figure label says ‘No Helm.’

Line 334-338: the data shows no significant difference between the modified helmet and the bare head condition as well as the conventional helmet. The reason for that seems to be a wider spread of the font-rear confusion rates between test subjects. The spread even exceeds that of the conventional helmet condition. For most of the test subjects there might be a benefit, but it should be discussed why some of the test subjects performed worse with the modified helmet. There is a section at the end of the discussion which mentions the fact, that the fit of the helmets could have been not tight enough to block the direct sound. However, this should be a benefit for the localization accuracy.

Line 360-… : If the goal was to see if the new ear pads could compensate for the attenuation of conventional ear pads with regards to localization it would be interesting to measure the hearing thresholds from different directions or use a dummy head to get the frequency response at different angles. The last approach could also be used to see why the modified sound did not yield better results.

Line 379-… : the attenuated frequencies were just measured for 0° and then used for 360° presentation. There is no discussion if the attenuation is the same for every angle.

Line 426: The authors seem to be involved in designing the ear pads and have no interest in sharing/describing it. If the ear pads are planned for distribution and commercial use, there might be a conflict of interest here.

Author Response

Dear Reviewer3

My replies see attached document

Kind regards,

Josef Seebacher

Round 2

Author Response

see attached document

Reviewer 3 Report

I think the manuscript improved with the revisions but I still miss a schematic drawing of the new ear pads.

L21-22: I don’t see enough evidence for this statement by this study. The only statement which should be made here based on the results of this study, is that the audiometric thresholds improve with the new ear pads.

L59: “give“ should be written with the letter “s“ here.

L95-96: A reference to Blauert should be provided.

L177: The statement that wholes in the ear pad did not improve spatial hearing should ideally be accompanied by test results.

Figure 3: Was there a statistical difference between sound localization results with the commercial versus new ear pads? Please add this information to the Figure.

Author Response

see attached document
